# Matrix factorization with Binary Components

**Martin Slawski, Matthias Hein and Pavlo Lutsik**
Saarland University
{ms,hein}@cs.uni-saarland.de,p.lutsik@mx.uni-saarland.de

## Abstract

Motivated by an application in computational biology, we consider low-rank matrix factorization with $\{0,1\}$-constraints on one of the factors and optionally convex constraints on the second one. In addition to the non-convexity shared with other matrix factorization schemes, our problem is further complicated by a combinatorial constraint set of size $2^{m \cdot r}$, where $m$ is the dimension of the data points and $r$ the rank of the factorization. Despite apparent intractability, we provide $-$ in the line of recent work on non-negative matrix factorization by Arora et al. (2012)$-$ an algorithm that provably recovers the underlying factorization in the exact case with $O(mr2^r + mnr + r^2 n)$ operations for $n$ datapoints. To obtain this result, we use theory around the Littlewood-Offord lemma from combinatorics.

## 1 Introduction

Low-rank matrix factorization techniques like the singular value decomposition (SVD) constitute an important tool in data analysis yielding a compact representation of data points as linear combinations of a comparatively small number of 'basis elements' commonly referred to as *factors*, *components* or *latent variables*. Depending on the specific application, the basis elements may be required to fulfill additional properties, e.g. non-negativity [1, 2], smoothness [3] or sparsity [4, 5]. In the present paper, we consider the case in which the basis elements are constrained to be binary, i.e. we aim at factorizing a real-valued data matrix $D$ into a product $TA$ with $T \in \{0,1\}^{m \times r}$ and $A \in \mathbb{R}^{r \times n}$, $r \ll \min\{m,n\}$. Such decomposition arises e.g. in blind source separation in wireless communication with binary source signals [6]; in network inference from gene expression data [7, 8], where $T$ encodes connectivity of transcription factors and genes; in unmixing of cell mixtures from DNA methylation signatures [9] in which case $T$ represents presence/absence of methylation; or in clustering with overlapping clusters with $T$ as a matrix of cluster assignments [10, 11].

Several other matrix factorizations involving binary matrices have been proposed in the literature. In [12] and [13] matrix factorization for binary input data, but non-binary factors $T$ and $A$ is discussed, whereas a factorization $TWA$ with both $T$ and $A$ binary and real-valued $W$ is proposed in [14], which is more restrictive than the model of the present paper. The model in [14] in turn encompasses binary matrix factorization as proposed in [15], where all of $D$, $T$ and $A$ are constrained to be binary. It is important to note that this ine of research is fundamentally different from Boolean matrix factorization [16], which is sometimes also referred to as binary matrix factorization.

A major drawback of matrix factorization schemes is non-convexity. As a result, there is in general no algorithm that is guaranteed to compute the desired factorization. Algorithms such as block coordinate descent, EM, MCMC, etc. commonly employed in practice lack theoretical guarantees beyond convergence to a local minimum. Substantial progress in this regard has been achieved recently for non-negative matrix factorization (NMF) by Arora et al. [17] and follow-up work in [18], where it is shown that under certain additional conditions, the NMF problem can be solved globally optimal by means of linear programming. Apart from being a non-convex problem, the matrix factorization studied in the present paper is further complicated by the $\{0,1\}$-constraints imposed on the left factor $T$, which yields a combinatorial optimization problem that appears to be computationally intractable except for tiny dimensions $m$ and $r$ even in case the right factor $A$ were

already known. Despite the obvious hardness of the problem, we present as our main contribution an algorithm that provably provides an exact factorization $D = TA$ whenever such factorization exists. Our algorithm has exponential complexity only in the rank $r$ of the factorization, but scales linearly in $m$ and $n$. In particular, the problem remains tractable even for large values of $m$ as long as $r$ remains small. We extend the algorithm to the approximate case $D \approx TA$ and empirically show superior performance relative to heuristic approaches to the problem. Moreover, we establish uniqueness of the exact factorization under the separability condition from the NMF literature [17, 19], or alternatively with high probability for $T$ drawn uniformly at random. As a corollary, we obtain that at least for these two models, the suggested algorithm continues to be fully applicable if additional constraints e.g. non-negativity, are imposed on the right factor $A$. We demonstrate the practical usefulness of our approach in unmixing DNA methylation signatures of blood samples [9].
**Notation.** For a matrix $M$ and index sets $I, J$, $M_{I,J}$ denotes the submatrix corresponding to $I$ and $J$; $M_{I,:}$ and $M_{:,J}$ denote the submatrices formed by the rows in $I$ respectively columns in $J$. We write $[M; M']$ and $[M, M']$ for the row- respectively column-wise concatenation of $M$ and $M'$. The affine hull generated by the columns of $M$ is denoted by $\mathrm{aff}(M)$. The symbols $\mathbf{1}/0$ denote vectors or matrices of ones/zeroes and $I$ denotes the identity matrix. We use $|\cdot|$ for the cardinality of a set.
**Supplement.** The supplement contains all proofs, additional comments and experimental results.

## 2 Exact case

We start by considering the exact case, i.e. we suppose that a factorization having the desired properties exists. We first discuss the geometric ideas underlying our basic approach for recovering such factorization from the data matrix before presenting conditions under which the factorization is unique. It is shown that the question of uniqueness as well as the computational performance of our approach is intimately connected to the Littlewood-Offord problem in combinatorics [20].

**2.1 Problem formulation.** Given $D \in \mathbb{R}^{m \times n}$, we consider the following problem.

$$\text{find } T \in \{0,1\}^{m \times r} \text{ and } A \in \mathbb{R}^{r \times n}, A^\top \mathbf{1}_r = \mathbf{1}_n \text{ such that } D = TA. \tag{1}$$

The columns $\{T_{:,k}\}_{k=1}^r$ of $T$, which are vertices of the hypercube $[0,1]^m$, are referred to as components. The requirement $A^\top \mathbf{1}_r = \mathbf{1}_n$ entails that the columns of $D$ are affine instead of linear combinations of the columns of $T$. This additional constraint is not essential to our approach; it is imposed for reasons of presentation, in order to avoid that the origin is treated differently from the other vertices of $[0,1]^m$, because otherwise the zero vector could be dropped from $T$, leaving the factorization unchanged. We further assume w.l.o.g. that $r$ is minimal, i.e. there is no factorization of the form (1) with $r' < r$, and in turn that the columns of $T$ are affinely independent, i.e. $\forall \lambda \in \mathbb{R}^r, \lambda^\top \mathbf{1}_r = 0$, $T\lambda = 0$ implies that $\lambda = 0$. Moreover, it is assumed that $\mathrm{rank}(A) = r$. This ensures the existence of a submatrix $A_{:,\mathcal{C}}$ of $r$ linearly independent columns and of a corresponding submatrix of $D_{:,\mathcal{C}}$ of affinely independent columns, when combined with the affine independence of the columns of $T$:

$$\forall \lambda \in \mathbb{R}^r, \lambda^\top \mathbf{1}_r = 0: D_{:,\mathcal{C}}\lambda = 0 \iff T(A_{:,\mathcal{C}}\lambda) = 0 \implies A_{:,\mathcal{C}}\lambda = 0 \implies \lambda = 0, \tag{2}$$

using at the second step that $\mathbf{1}_r^\top A_{:,\mathcal{C}}\lambda = \mathbf{1}_r^\top \lambda = 0$ and the affine independence of the $\{T_{:,k}\}_{k=1}^r$. Note that the assumption $\mathrm{rank}(A) = r$ is natural; otherwise, the data would reside in an affine subspace of lower dimension so that $D$ would not contain enough information to reconstruct $T$.

**2.2 Approach.** Property (2) already provides the entry point of our approach. From $D = TA$, it is obvious that $\mathrm{aff}(T) \supseteq \mathrm{aff}(D)$. Since $D$ contains the same number of affinely independent columns as $T$, it must also hold that $\mathrm{aff}(D) \supseteq \mathrm{aff}(T)$, in particular $\mathrm{aff}(D) \supseteq \{T_{:,k}\}_{k=1}^r$. Consequently, (1) can in principle be solved by enumerating all vertices of $[0,1]^m$ contained in $\mathrm{aff}(D)$ and selecting a maximal affinely independent subset thereof (see Figure 1). This procedure, however, is exponential in the dimension $m$, with $2^m$ vertices to be checked for containment in $\mathrm{aff}(D)$ by solving a linear system. Remarkably, the following observation along with its proof, which prompts Algorithm 1 below, shows that the number of elements to be checked can be reduced to $2^{r-1}$ irrespective of $m$.

**Proposition 1.** *The affine subspace $\mathrm{aff}(D)$ contains no more than $2^{r-1}$ vertices of $[0,1]^m$. Moreover, Algorithm 1 provides all vertices contained in $\mathrm{aff}(D)$.*

---

**Algorithm 1** FINDVERTICES EXACT

1. Fix $p \in \mathrm{aff}(D)$ and compute $P = [D_{:,1} - p, \ldots, D_{:,n} - p]$.
2. Determine $r - 1$ linearly independent columns $\mathcal{C}$ of $P$, obtaining $P_{:,\mathcal{C}}$ and subsequently $r - 1$ linearly independent rows $\mathcal{R}$, obtaining $P_{\mathcal{R},\mathcal{C}} \in \mathbb{R}^{r-1 \times r-1}$.
3. Form $Z = P_{:,\mathcal{C}}(P_{\mathcal{R},\mathcal{C}})^{-1} \in \mathbb{R}^{m \times r-1}$ and $\widehat{T} = Z(B^{(r-1)} - p_{\mathcal{R}} \mathbf{1}_{2^{r-1}}^\top) + p\mathbf{1}_{2^{r-1}}^\top \in \mathbb{R}^{m \times 2^{r-1}}$, where the columns of $B^{(r-1)}$ correspond to the elements of $\{0,1\}^{r-1}$.
4. Set $\mathcal{T} = \emptyset$. For $u = 1, \ldots, 2^{r-1}$, if $\widehat{T}_{:,u} \in \{0,1\}^m$ set $\mathcal{T} = \mathcal{T} \cup \{\widehat{T}_{:,u}\}$.
5. Return $\mathcal{T} = \{0,1\}^m \cap \mathrm{aff}(D)$.

---

**Algorithm 2** BINARYFACTORIZATION EXACT

1. Obtain $\mathcal{T}$ as output from FINDVERTICES EXACT($D$)
2. Select $r$ affinely independent elements of $\mathcal{T}$ to be used as columns of $T$.
3. Obtain $A$ as solution of the linear system $[\mathbf{1}_r^\top; T]A = [\mathbf{1}_n^\top; D]$.
4. Return $(T, A)$ solving problem (1).

---

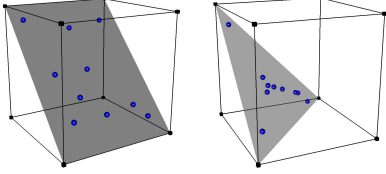

Figure 1: Illustration of the geometry underlying our approach in dimension $m = 3$. Dots represent data points and the shaded areas their affine hulls $\mathrm{aff}(D) \cap [0,1]^m$. Left: $\mathrm{aff}(D)$ intersects with $r + 1$ vertices of $[0,1]^m$. Right: $\mathrm{aff}(D)$ intersects with precisely $r$ vertices.

**Comments.** In step 2 of Algorithm 1, determining the rank of $P$ and an associated set of linearly independent columns/rows can be done by means of a rank-revealing QR factorization [21, 22]. The crucial step is the third one, which is a compact description of first solving the linear systems $P_{\mathcal{R},\mathcal{C}}\lambda = b - p_{\mathcal{R}}$ for all $b \in \{0,1\}^{r-1}$ and back-substituting the result to compute candidate vertices $P_{:,\mathcal{C}}\lambda + p$ stacked into the columns of $\widehat{T}$; the addition/subtraction of $p$ is merely because we have to deal with an affine instead of a linear subspace, in which $p$ serves as origin. In step 4, the pool of $2^{r-1}$ 'candidates' is filtered, yielding $\mathcal{T} = \mathrm{aff}(D) \cap \{0,1\}^m$.

Determining $\mathcal{T}$ is the hardest part in solving the matrix factorization problem (1). Given $\mathcal{T}$, the solution can be obtained after few inexpensive standard operations. Note that step 2 in Algorithm 2 is not necessary if one does not aim at finding a minimal factorization, i.e. if it suffices to have $D = TA$ with $T \in \{0,1\}^{m \times r'}$ but $r'$ possibly being larger than $r$.

As detailed in the supplement, the case without sum-to-one constraints on $A$ can be handled similarly, as can be the model in [14] with binary left and right factor and real-valued middle factor.

**Computational complexity.** The dominating cost in Algorithm 1 is computation of the candidate matrix $\widehat{T}$ and checking whether its columns are vertices of $[0,1]^m$. Note that

$$\widehat{T}_{\mathcal{R},:} = Z_{\mathcal{R},:}(B^{(r-1)} - p_{\mathcal{R}} \mathbf{1}_{2^{r-1}}^\top) + p_{\mathcal{R}}\mathbf{1}_{2^{r-1}}^\top = I_{r-1}(B^{(r-1)} - p_{\mathcal{R}}\mathbf{1}_{2^{r-1}}^\top) + p_{\mathcal{R}}\mathbf{1}_{2^{r-1}}^\top = B^{(r-1)}, \quad (3)$$

i.e. the $r - 1$ rows of $\widehat{T}$ corresponding to $\mathcal{R}$ do not need to be taken into account. Forming the matrix $\widehat{T}$ would hence require $O((m - r + 1)(r - 1)2^{r-1})$ and the subsequent check for vertices in the fourth step $O((m - r + 1)2^{r-1})$ operations. All other operations are of lower order provided e.g. $(m - r + 1)2^{r-1} > n$. The second most expensive operation is forming the matrix $P_{\mathcal{R},\mathcal{C}}$ in step 2 with the help of a QR decomposition requiring $O(mn(r - 1))$ operations in typical cases [21]. Computing the matrix factorization (1) after the vertices have been identified (steps 2 to 4 in Algorithm 2) has complexity $O(mnr + r^3 + r^2n)$. Here, the dominating part is the solution of a linear system in $r$ variables and $n$ right hand sides. Altogether, our approach for solving (1) has exponential complexity in $r$, but only linear complexity in $m$ and $n$. Later on, we will argue that under additional assumptions on $T$, the $O((m-r+1)2^{r-1})$ terms can be reduced to $O((r-1)2^{r-1})$.

**2.3 Uniqueness.** In this section, we study uniqueness of the matrix factorization problem (1) (modulo permutation of columns/rows). First note that in view of the affine independence of the columns of $T$, the factorization is unique iff $T$ is, which holds iff

$$\mathrm{aff}(D) \cap \{0,1\}^m = \mathrm{aff}(T) \cap \{0,1\}^m = \{T_{:,1}, \ldots, T_{:,r}\}, \quad (4)$$

i.e. if the affine subspace generated by $\{T_{:,1}, \ldots, T_{:,r}\}$ contains no other vertices of $[0,1]^m$ than the $r$ given ones (cf. Figure 1). Uniqueness is of great importance in applications, where one aims at

an interpretation in which the columns of $T$ play the role of underlying data-generating elements. Such an interpretation is not valid if (4) fails to hold, since it is then possible to replace one of the columns of a specific choice of $T$ by another vertex contained in the same affine subspace.

**Solution of a non-negative variant of our factorization.** In the sequel, we argue that property (4) plays an important role from a computational point of view when solving extensions of problem (1) in which further constraints are imposed on $A$. One particularly important extension is the following.

$$\text{find } T \in \{0,1\}^{m \times r} \text{ and } A \in \mathbb{R}_+^{r \times n}, A^\top \mathbf{1}_r = \mathbf{1}_n \text{ such that } D = TA. \tag{5}$$

Problem (5) is a special instance of non-negative matrix factorization. Problem (5) is of particular interest in the present paper, leading to a novel real world application of matrix factorization techniques as presented in Section 4.2 below. It is natural to ask whether Algorithm 2 can be adapted to solve problem (5). A change is obviously required for the second step when selecting $r$ vertices from $\mathcal{T}$, since in (5) the columns $D$ now have to be expressed as convex instead of only affine combinations of columns of $T$: picking an affinely independent collection from $\mathcal{T}$ does not take into account the non-negativity constraint imposed on $A$. If, however, (4) holds, we have $|\mathcal{T}| = r$ and Algorithm 2 must return a solution of (5) provided that there exists one.

**Corollary 1.** *If problem* (1) *has a unique solution, i.e. if condition* (4) *holds and if there exists a solution of* (5)*, then it is returned by Algorithm 2.*

To appreciate that result, consider the converse case $|\mathcal{T}| > r$. Since the aim is a minimal factorization, one has to find a subset of $\mathcal{T}$ of cardinality $r$ such that (5) can be solved. In principle, this can be achieved by solving a linear program for $\binom{|\mathcal{T}|}{r}$ subsets of $\mathcal{T}$, but this is in general not computationally feasible: the upper bound of Proposition 1 indicates that $|\mathcal{T}| = 2^{r-1}$ in the worst case. For the example below, $\mathcal{T}$ consists of all $2^{r-1}$ vertices contained in an $r-1$-dimensional face of $[0,1]^m$:

$$T = \begin{pmatrix} 0_{m-r \times r} \\ I_{r-1} \ 0_{r-1} \\ 0_r^\top \end{pmatrix} \quad \text{with } \mathcal{T} = \left\{ T\lambda : \lambda_1 \in \{0,1\}, \dots, \lambda_{r-1} \in \{0,1\}, \lambda_r = 1 - \sum_{k=1}^{r-1} \lambda_k \right\}. \tag{6}$$

**Uniqueness under separability.** In view of the negative example (6), one might ask whether uniqueness according to (4) can be achieved under additional conditions on $T$. We prove uniqueness under *separability*, a condition introduced in [19] and imposed recently in [17] to show solvability of the NMF problem by linear programming. We say that $T$ is separable if there exists a permutation $\Pi$ such that $\Pi T = [M; I_r]$, where $M \in \{0,1\}^{m-r \times r}$.

**Proposition 2.** *If $T$ is separable, condition* (4) *holds and thus problem* (1) *has a unique solution.*

**Uniqueness under generic random sampling.** Both the negative example (6) as well as the positive result of Proposition 2 are associated with special matrices $T$. This raises the question whether uniqueness holds respectively fails for broader classes of binary matrices. In order to gain insight into this question, we consider random $T$ with i.i.d. entries from a Bernoulli distribution with parameter $\frac{1}{2}$ and study the probability of the event $\{\text{aff}(T) \cap \{0,1\}^m = \{T_{:,1}, \dots, T_{:,r}\}\}$. This question has essentially been studied in combinatorics [23], with further improvements in [24]. The results therein rely crucially on Littlewood-Offord theory (see Section 2.4 below).

**Theorem 1.** *Let $T$ be a random $m \times r$-matrix whose entries are drawn i.i.d. from $\{0,1\}$ with probability $\frac{1}{2}$. Then, there is a constant $C$ so that if $r \le m - C$,*

$$\mathbf{P}\left(\text{aff}(T) \cap \{0,1\}^m = \{T_{:,1}, \dots, T_{:,r}\}\right) \ge 1 - (1+o(1)) \, 4\binom{r}{3}\left(\frac{3}{4}\right)^m - \left(\frac{3}{4} + o(1)\right)^m \quad \text{as } m \to \infty.$$

Theorem 1 suggests a positive answer to the question of uniqueness posed above. For $m$ large enough and $r$ small compared to $m$ (in fact, following [24] one may conjecture that Theorem 1 holds with $C = 1$), the probability that the affine hull of $r$ vertices of $[0,1]^m$ selected uniformly at random contains some other vertex is exponentially small in the dimension $m$. We have empirical evidence that the result of Theorem 1 continues to hold if the entries of $T$ are drawn from a Bernoulli distribution with parameter in $(0,1)$ sufficiently far away from the boundary points (cf. supplement). As a byproduct, these results imply that also the NMF variant of our matrix factorization problem (5) can in most cases be reduced to identifying a set of $r$ vertices of $[0,1]^m$ (cf. Corollary 1).

**2.4 Speeding up Algorithm 1.** In Algorithm 1, an $m \times 2^{r-1}$ matrix $\widehat{T}$ of potential vertices is formed (Step 3). We have discussed the case (6) where all candidates must indeed be vertices, in which case it seems to be impossible to reduce the computational cost of $O((m-r)r2^{r-1})$, which becomes significant once $m$ is in the thousands and $r \geq 25$. On the positive side, Theorem 1 indicates that for many instances of $T$, only $r$ out of $2^{r-1}$ candidates are in fact vertices. In that case, noting that columns of $\widehat{T}$ cannot be vertices if a single coordinate is not in $\{0,1\}$ (and that the vast majority of columns of $\widehat{T}$ must have one such coordinate), it is computationally more favourable to incrementally compute subsets of rows of $\widehat{T}$ and then to discard already those columns with coordinates not in $\{0,1\}$. We have observed empirically that this scheme rapidly reduces the candidate set $-$ already checking a single row of $\widehat{T}$ eliminates a substantial portion (see Figure 2).

**Littlewood-Offord theory.** Theoretical underpinning for the last observation can be obtained from a result in combinatorics, the *Littlewood-Offord* (L-O)-lemma. Various extensions of that result have been developed until recently, see the survey [25]. We here cite the L-O-lemma in its basic form.

**Theorem 2.** *[20] Let $a_1, \ldots, a_\ell \in \mathbb{R} \setminus \{0\}$ and $y \in \mathbb{R}$.*

    *(i)* $\left| \{ b \in \{0,1\}^\ell : \sum_{i=1}^\ell a_i b_i = y \} \right| \leq \binom{\ell}{\lfloor \ell/2 \rfloor}$.

    *(ii) If $|a_i| \geq 1$, $i = 1, \ldots, \ell$, $\left| \{ b \in \{0,1\}^\ell : \sum_{i=1}^\ell a_i b_i \in (y, y+1) \} \right| \leq \binom{\ell}{\lfloor \ell/2 \rfloor}$.*

The two parts of Theorem 2 are referred to as discrete respectively continuous L-O lemma. The discrete L-O lemma provides an upper bound on the number of $\{0,1\}$-vectors whose weighted sum with given weights $\{a_i\}_{i=1}^\ell$ is equal to some given number $y$, whereas the stronger continuous version, under a more stringent condition on the weights, upper bounds the number of $\{0,1\}$-vectors whose weighted sum is contained in some interval $(y, y+1)$. In order to see the relation of Theorem 2 to Algorithm 1, let us re-inspect the third step of that algorithm. To obtain a reduction of candidates by checking a single row of $\widehat{T} = Z(B^{(r-1)} - p_{\mathcal{R}} \mathbf{1}_{2^{r-1}}^\top) + p \mathbf{1}_{2^{r-1}}^\top$, pick $i \notin \mathcal{R}$ (recall that coordinates in $\mathcal{R}$ do not need to be checked, cf. (3)) and $u \in \{1, \ldots, 2^{r-1}\}$ arbitrary. The $u$-th candidate can be a vertex only if $\widehat{T}_{i,u} \in \{0,1\}$. The condition $\widehat{T}_{i,u} = 0$ can be written as

$$\underbrace{Z_{i,:}}_{\{a_k\}_{k=1}^r} \underbrace{B_{:,u}^{(r-1)}}_{=b} = \underbrace{Z_{i,:} p_{\mathcal{R}} - p_i}_{=y}. \tag{7}$$

A similar reasoning applies when setting $\widehat{T}_{i,u} = 1$. Provided none of the entries of $Z_{i,:} = 0$, the discrete L-O lemma implies that there are at most $2\binom{r-1}{\lfloor (r-1)/2 \rfloor}$ out of $2^{r-1}$ candidates for which the $i$-th coordinate is in $\{0,1\}$. This yields a reduction of the candidate set by $2\binom{r-1}{\lfloor (r-1)/2 \rfloor}/2^{r-1} = O\left(\frac{1}{\sqrt{r-1}}\right)$. Admittedly, this reduction may appear insignificant given the total number of candidates to be checked. The reduction achieved empirically (cf. Figure 2) is typically larger. Stronger reductions have been proven under additional assumptions on the weights $\{a_i\}_{i=1}^\ell$: e.g. for distinct weights, one obtains a reduction of $O((r-1)^{-3/2})$ [25]. Furthermore, when picking successively $d$ rows of $\widehat{T}$ and if one assumes that each row yields a reduction according to the discrete L-O lemma, one would obtain the reduction $(r-1)^{-d/2}$ so that $d = r - 1$ would suffice to identify all vertices provided $r \geq 4$. Evidence for the rate $(r-1)^{-d/2}$ can be found in [26]. This indicates a reduction in complexity of Algorithm 1 from $O((m-r)r2^{r-1})$ to $O(r^2 2^{r-1})$.

**Achieving further speed-up with integer linear programming.** The continuous L-O lemma (part (ii) of Theorem 2) combined with the derivation leading to (7) allows us to tackle even the case $r = 80$ ($2^{80} \approx 10^{24}$). In view of the continuous L-O lemma, a reduction in the number of candidates can still be achieved if the requirement is weakened to $\widehat{T}_{i,u} \in [0,1]$. According to (7) the candidates satisfying the relaxed constraint for the $i$-th coordinate can be obtained from the feasibility problem

$$\text{find } b \in \{0,1\}^{r-1} \text{ subject to } 0 \leq Z_{i,:}(b - p_{\mathcal{R}}) + p_i \leq 1, \tag{8}$$

which is an integer linear program that can be solved e.g. by CPLEX. The L-O- theory suggests that the branch-bound strategy employed therein is likely to be successful. With the help of CPLEX, it is affordable to solve problem (8) with all $m - r + 1$ constraints (one for each of the rows of $\widehat{T}$ to be checked) imposed simultaneously. We always recovered directly the underlying vertices in our experiments and only these, without the need to prune the solution pool (which could be achieved by Algorithm 1, replacing the $2^{r-1}$ candidates by a potentially much smaller solution pool).

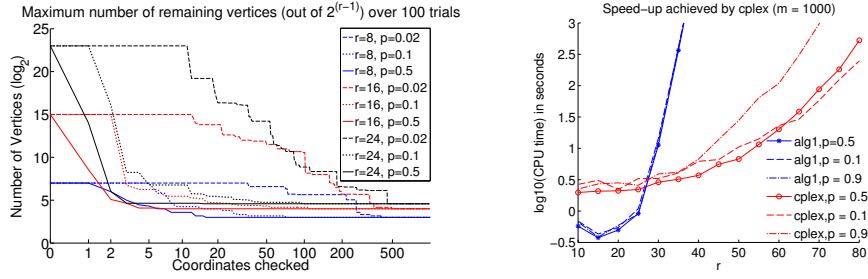

Figure 2: Left: Speeding up the algorithm by checking single coordinates, remaining number of coordinates vs.# coordinates checked ($m = 1000$). Right: Speed up by CPLEX compared to Algorithm 1. For both plots, $T$ is drawn entry-wise from a Bernoulli distribution with parameter $p$.

## 3 Approximate case

In the sequel, we discuss an extension of our approach to handle the approximate case $D \approx TA$ with $T$ and $A$ as in (1). In particular, we have in mind the case of additive noise i.e. $D = TA + E$ with $\|E\|_F$ small. While the basic concept of Algorithm 1 can be adopted, changes are necessary because $D$ may have full rank $\min\{m, n\}$ and second aff$(D) \cap \{0,1\}^m = \emptyset$, i.e. the distances of aff$(D)$ and the $\{T_{:,k}\}_{k=1}^r$ may be strictly positive (but are at least assumed to be small). As dis-

---

**Algorithm 3** FINDVERTICES APPROXIMATE

---

1. Let $p = D\mathbf{1}_n/n$ and compute $P = [D_{:,1} - p, \ldots, D_{:,n} - p]$.
2. Compute $U^{(r-1)} \in \mathbb{R}^{m \times r-1}$, the left singular vectors corresponding to the $r - 1$ largest singular values of $P$. Select $r - 1$ linearly independent rows $\mathcal{R}$ of $U^{(r-1)}$, obtaining $U^{(r-1)}_{\mathcal{R},:} \in \mathbb{R}^{r-1 \times r-1}$.
3. Form $Z = U^{(r-1)}(U^{(r-1)}_{\mathcal{R},:})^{-1}$ and $\widehat{T} = Z(B^{(r-1)} - p_{\mathcal{R}}\mathbf{1}_{2^{r-1}}^{\top}) + p\mathbf{1}_{2^{r-1}}^{\top}$.
4. Compute $\widehat{T}^{01} \in \mathbb{R}^{m \times 2^{r-1}}$: for $u = 1, \ldots, 2^{r-1}$, $i = 1, \ldots, m$, set $\widehat{T}^{01}_{i,u} = I(\widehat{T}_{i,u} > \frac{1}{2})$.
5. For $u = 1, \ldots, 2^{r-1}$, set $\delta_u = \|\widehat{T}_{:,u} - \widehat{T}^{01}_{:,u}\|_2$. Order increasingly s.t. $\delta_{u_1} \leq \ldots \leq \delta_{2^{r-1}}$.
6. Return $T = [\widehat{T}^{01}_{:,u_1} \ldots \widehat{T}^{01}_{:,u_r}]$

---

tinguished from the exact case, Algorithm 3 requires the number of components $r$ to be specified in advance as it is typically the case in noisy matrix factorization problems. Moreover, the vector $p$ subtracted from all columns of $D$ in step 1 is chosen as the mean of the data points, which is in particular a reasonable choice if $D$ is contaminated with additive noise distributed symmetrically around zero. The truncated SVD of step 2 achieves the desired dimension reduction and potentially reduces noise corresponding to small singular values that are discarded. The last change arises in step 5. While in the exact case, one identifies all columns of $\widehat{T}$ that are in $\{0,1\}^m$, one instead only identifies columns close to $\{0,1\}^m$. Given the output of Algorithm 3, we solve the approximate matrix factorization problem via least squares, obtaining the right factor from $\min_A \|D - TA\|_F^2$.

**Refinements.** Improved performance for higher noise levels can be achieved by running Algorithm 3 multiple times with different sets of rows selected in step 2, which yields candidate matrices $\{T^{(l)}\}_{l=1}^s$, and subsequently using $T = \operatorname{argmin}_{\{T^{(l)}\}} \min_A \|D - T^{(l)}A\|_F^2$, i.e. one picks the candidate yielding the best fit. Alternatively, we may form a candidate pool by merging the $\{T^{(l)}\}_{l=1}^s$ and then use a backward elimination scheme, in which successively candidates are dropped that yield the smallest improvement in fitting $D$ until $r$ candidates are left. Apart from that, $T$ returned by Algorithm 3 can be used for initializing the block optimization scheme of Algorithm 4 below. Algorithm 4 is akin to standard block coordinate descent schemes proposed in the matrix factorization literature, e.g. [27]. An important observation (step 3) is that optimization of $T$ is separable along the rows of $T$, so that for small $r$, it is feasible to perform exhaustive search over all $2^r$ possibilities (or to use CPLEX). However, Algorithm 4 is impractical as a stand-alone scheme, because without proper initialization, it may take many iterations to converge, with each single iteration being more expensive than Algorithm 3. When initialized with the output of the latter, however, we have observed convergence of the block scheme only after few steps.

**Algorithm 4** Block optimization scheme for solving $\min_{T\in\{0,1\}^{m\times r},\, A}\|D-TA\|_F^2$

1. Set $k=0$ and set $T^{(k)}$ equal to a starting value.
2. $A^{(k)} \leftarrow \operatorname{argmin}_A \|D-T^{(k)}A\|_F^2$ and set $k=k+1$.
3. $T^{(k)} \leftarrow \operatorname{argmin}_{T\in\{0,1\}^{m\times r}}\|D-TA^{(k)}\|_F^2 = \operatorname{argmin}_{\{T_{i,:}\in\{0,1\}^r\}_{i=1}^m}\sum_{i=1}^m\|D_{i,:}-T_{i,:}A^{(k)}\|_2^2$ (9)
4. Alternate between steps 2 and 3.

## 4 Experiments

In Section 4.1 we demonstrate with the help of synthetic data that the approach of Section 3 performs well on noisy datasets. In the second part, we present an application to a real dataset.

### 4.1 Synthetic data.

**Setup.** We generate $D = T^*A^* + \alpha E$, where the entries of $T^*$ are drawn i.i.d. from $\{0,1\}$ with probability 0.5, the columns of $A$ are drawn i.i.d. uniformly from the probability simplex and the entries of $E$ are i.i.d. standard Gaussian. We let $m = 1000$, $r = 10$ and $n = 2r$ and let the noise level $\alpha$ vary along a grid starting from 0. Small sample sizes $n$ as considered here yield more challenging problems and are motivated by the real world application of the next subsection.

**Evaluation.** Each setup is run 20 times and we report averages over the following performance measures: the normalized Hamming distance $\|T^* - T\|_F^2/(m\,r)$ and the two RMSEs $\|T^*A^* - TA\|_F/(m\,n)^{1/2}$ and $\|TA - D\|_F/(m\,n)^{1/2}$, where $(T, A)$ denotes the output of one of the following approaches that are compared. FindVertices: our approach in Section 3. oracle: we solve problem (9) with $A^{(k)} = A^*$. box: we run the block scheme of Algorithm 4, relaxing the integer constraint into a box constraint. Five random initializations are used and we take the result yielding the best fit, subsequently rounding the entries of $T$ to fulfill the $\{0,1\}$-constraints and refitting $A$. quad pen: as box, but a (concave) quadratic penalty $\lambda\sum_{i,k}T_{i,k}(1-T_{i,k})$ is added to push the entries of $T$ towards $\{0,1\}$. D.C. programming [28] is used for the block updates of $T$.

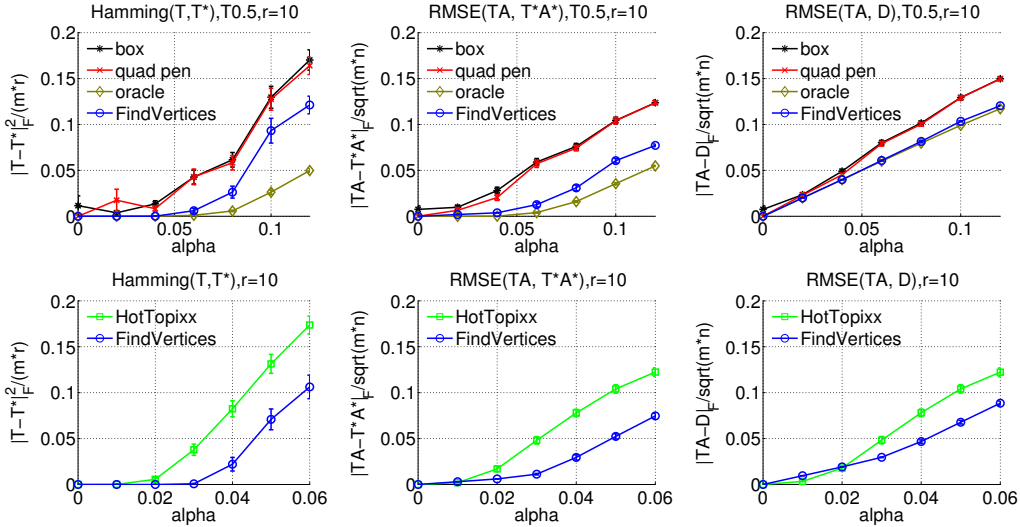

Figure 3: Top: comparison against block schemes. Bottom: comparison against HOTTOPIXX. Left/Middle/Right: $\|T^* - T\|_F^2/(m\,r)$, $\|T^*A^* - TA\|_F/(m\,n)^{1/2}$ and $\|TA - D\|_F/(m\,n)^{1/2}$.

**Comparison to HOTTOPIXX [18].** HOTTOPIXX (HT) is a linear programming approach to NMF equipped with guarantees such as correctness in the exact and robustness in the non-exact case as long as $T$ is (nearly) separable (cf. Section 2.3). HT does not require $T$ to be binary, but applies to the generic NMF problem $D \approx TA$, $T \in \mathbb{R}_+^{m\times r}$ and $A \in \mathbb{R}_+^{r\times n}$. Since separability is crucial to the performance of HT, we restrict our comparison to separable $T = [M; I_r]$, generating the entries of $M$ i.i.d. from a Bernoulli distribution with parameter 0.5. For runtime reasons, we lower the dimension to $m = 100$. Apart from that, the experimental setup is as above. We

use an implementation of HT from [29]. We first pre-normalize $D$ to have unit row sums as required by HT, and obtain $A$ as first output. Given $A$, the non-negative least squares problem $\min_{T \in \mathbb{R}_+^{m \times r}} \|D - TA\|_F^2$ is solved. The entries of $T$ are then re-scaled to match the original scale of $D$, and thresholding at 0.5 is applied to obtain a binary matrix. Finally, $A$ is re-optimized by solving the above fitting problem with respect to $A$ in place of $T$. In the noisy case, HT needs a tuning parameter to be specified that depends on the noise level, and we consider a grid of 12 values for that parameter. The range of the grid is chosen based on knowledge of the noise matrix $E$. For each run, we pick the parameter that yields best performance in favour of HT.

**Results.** From Figure 3, we find that unlike the other approaches, box does not always recover $T^*$ even if the noise level $\alpha = 0$. FindVertices outperforms box and quad pen throughout. For $\alpha \leq 0.06$, its performance closely matches that of the oracle. In the separable case, our approach performs favourably as compared to HT, a natural benchmark in this setting.

### 4.2 Analysis of DNA methylation data.
**Background.** Unmixing of DNA methylation profiles is a problem of high interest in cancer research. DNA methylation is a chemical modification of the DNA occurring at specific sites, so-called CpGs. DNA methylation affects gene expression and in turn various processes such as cellular differentiation. A site is either unmethylated ('0') or methylated ('1'). DNA methylation microarrays allow one to measure the methylation level for thousands of sites. In the dataset considered here, the measurements $D$ (the rows corresponding to sites, the columns to samples) result from a mixture of cell types. The methylation profiles of the latter are in $\{0,1\}^m$, whereas, depending on the mixture proportions associated with each sample, the entries of $D$ take values in $[0,1]^m$. In other words, we have the model $D \approx TA$, with $T$ representing the methylation of the cell types and the columns of $A$ being elements of the probability simplex. It is often of interest to recover the mixture proportions of the samples, because e.g. specific diseases, in particular cancer, can be associated with shifts in these proportions. The matrix $T$ is frequently unknown, and determining it experimentally is costly. Without $T$, however, recovering the mixing matrix $A$ is challenging, in particular since the number of samples in typical studies is small.

**Dataset.** We consider the dataset studied in [9], with $m = 500$ CpG sites and $n = 12$ samples of blood cells composed of four major types (B-/T-cells, granulocytes, monocytes), i.e. $r = 4$. Ground truth is partially available: the proportions of the samples, denoted by $A^*$, are known.

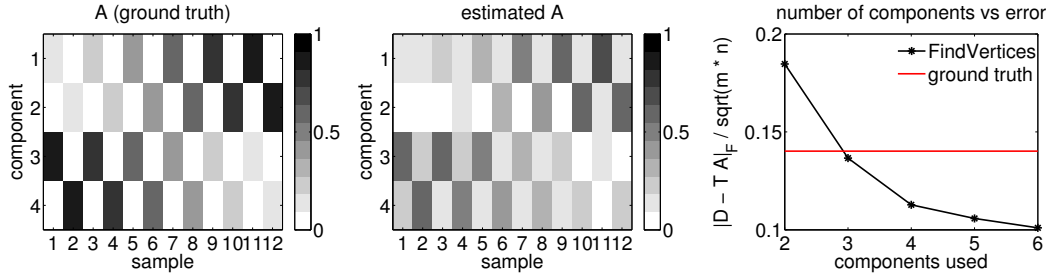

Figure 4: Left: Mixture proportions of the ground truth. Middle: mixture proportions as estimated by our method. Right: RMSEs $\|D - \overline{T}\,\overline{A}\|_F / (m\,n)^{1/2}$ in dependency of $r$.

**Analysis.** We apply our approach to obtain an approximate factorization $D \approx \overline{T}\,\overline{A}, \overline{T} \in \{0,1\}^{m \times r}$, $\overline{A} \in \mathbb{R}_+^{r \times n}$ and $\overline{A}^\top \mathbf{1}_r = \mathbf{1}_n$. We first obtained $\overline{T}$ as outlined in Section 3, replacing $\{0,1\}$ by $\{0.1, 0.9\}$ in order to account for measurement noise in $D$ that slightly pushes values towards 0.5. This can be accomodated re-scaling $\widehat{T}^{01}$ in step 4 of Algorithm 3 by 0.8 and then adding 0.1. Given $\overline{T}$, we solve the quadratic program $\overline{A} = \operatorname{argmin}_{A \in \mathbb{R}_+^{r \times n}, A^\top \mathbf{1}_r = \mathbf{1}_n} \|D - \overline{T}A\|_F^2$ and compare $\overline{A}$ to the ground truth $A^*$. In order to judge the fit as well as the matrix $\overline{T}$ returned by our method, we compute $T^* = \operatorname{argmin}_{T \in \{0,1\}^{m \times r}} \|D - TA^*\|_F^2$ as in (9). We obtain 0.025 as average mean squared difference of $\overline{T}$ and $T^*$, which corresponds to an agreement of 96 percent. Figure 4 indicates at least a qualitative agreement of $A^*$ and $\overline{A}$. In the rightmost plot, we compare the RMSEs of our approach for different choices of $r$ relative to the RMSE of $(T^*, A^*)$. The error curve flattens after $r = 4$, which suggests that with our approach, we can recover the correct number of cell types.

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
