[Supplementary Material · nips2013_supplement.pdf]

# Supplement to
# 'Matrix factorization with Binary Components'

**Martin Slawski, Matthias Hein and Pavlo Lutsik**
Saarland University
{ms,hein}@cs.uni-saarland.de,p.lutsik@mx.uni-saarland.de

The supplement contains all proofs, additional comments and experimental results.

## A Proof of Proposition 1

Proposition 1 is about Algorithm 1, which we re-state here.

---
**Algorithm 1** FINDVERTICES EXACT

1. Fix $p \in \text{aff}(D)$ and compute $P = [D_{:,1} - p, \ldots, D_{:,n} - p]$.
2. Determine $r-1$ linearly independent columns $\mathcal{C}$ of $P$, obtaining $P_{:,\mathcal{C}}$ and subsequently $r-1$ linearly independent rows $\mathcal{R}$, obtaining $P_{\mathcal{R},\mathcal{C}} \in \mathbb{R}^{r-1 \times r-1}$.
3. Form $Z = P_{:,\mathcal{C}}(P_{\mathcal{R},\mathcal{C}})^{-1} \in \mathbb{R}^{m \times r-1}$ and $\widehat{T} = Z(B^{(r-1)} - p_{\mathcal{R}} \mathbf{1}_{2^{r-1}}^{\top}) + p \mathbf{1}_{2^{r-1}}^{\top} \in \mathbb{R}^{m \times 2^{r-1}}$, where the columns of $B^{(r-1)}$ correspond to the elements of $\{0,1\}^{r-1}$.
4. Set $\mathcal{T} = \emptyset$. For $u = 1, \ldots, 2^{r-1}$, if $\widehat{T}_{:,u} \in \{0,1\}^m$ set $\mathcal{T} = \mathcal{T} \cup \{\widehat{T}_{:,u}\}$.
5. Return $\mathcal{T} = \{0,1\}^m \cap \text{aff}(D)$.

---

**Proposition 1.** *The affine subspace* $\text{aff}(D)$ *contains no more than* $2^{r-1}$ *vertices of* $[0,1]^m$. *Moreover, Algorithm 1 provides all vertices contained in* $\text{aff}(D)$.

*Proof.* Consider the first part of the statement. Let $b \in \{0,1\}^m$ and $p \in \text{aff}(D)$ arbitrary. We have $b \in \text{aff}(D)$ iff there exists $\theta \in \mathbb{R}^n$ s.t.

$$D\theta = b, \ \theta^{\top}\mathbf{1}_n = 1 \iff \underbrace{[D_{:,1} - p, \ldots, D_{:,n} - p]}_{=P}\theta + p = b \iff P\theta = b - p. \quad (A.1)$$

Note that $\text{rank}(P) = r - 1$. Hence, if there exists $\theta$ s.t. $P\theta = b - p$, such $\theta$ can be obtained from the unique $\lambda \in \mathbb{R}^{r-1}$ solving $P_{\mathcal{R},\mathcal{C}}\lambda = b_{\mathcal{R}} - p_{\mathcal{R}}$, where $\mathcal{R} \subset \{1,\ldots,m\}$ and $\mathcal{C} \subset \{1,\ldots,n\}$ are subsets of rows respectively columns of $P$ s.t. $\text{rank}(P_{\mathcal{R},\mathcal{C}}) = r-1$. Finally note that $b_{\mathcal{R}} \in \{0,1\}^{r-1}$ so that there are no more than $2^{r-1}$ distinct right hand sides $b_{\mathcal{R}} - p_{\mathcal{R}}$.

Turning to the second part of the statement, observe that for each $b \in \{0,1\}^m$, there exists a unique $\lambda$ s.t. $P_{\mathcal{R},\mathcal{C}}\lambda = b_{\mathcal{R}} - p_{\mathcal{R}} \Leftrightarrow \lambda = (P_{\mathcal{R},\mathcal{C}})^{-1}(b_{\mathcal{R}} - p_{\mathcal{R}})$. Repeating the argument preceding (A.1), if $b \in \{0,1\}^m \cap \text{aff}(D)$, it must hold that

$$b = P_{:,\mathcal{C}}\lambda + p \iff b = \underbrace{P_{:,\mathcal{C}}(P_{\mathcal{R},\mathcal{C}})^{-1}}_{=Z}(b_{\mathcal{R}} - p_{\mathcal{R}}) + p \iff b = Z(b_{\mathcal{R}} - p_{\mathcal{R}}) + p. \quad (A.2)$$

Algorithm 1 generates all possible right hand sides $\widehat{T} = Z(B^{(r-1)} - p_{\mathcal{R}}\mathbf{1}_{2^{r-1}}^{\top}) + p\mathbf{1}_{2^{r-1}}^{\top}$, where $B^{(r-1)}$ contains all elements of $\{0,1\}^{r-1}$ as its columns. Consequently if $b \in \{0,1\}^m \cap \text{aff}(D)$, it must appear as a column of $\widehat{T}$. Conversely, if the leftmost equality in (A.2) does not hold, $b \notin \text{aff}(D)$ and the column of $\widehat{T}$ corresponding to $b_{\mathcal{R}}$ cannot be a binary vector. $\square$

# B The matrix factorization problem without the constraint $A^\top \mathbf{1}_r = \mathbf{1}_n$

In the paper, we have provided Algorithm 2 to solve the matrix factorization problem

$$\text{find } T \in \{0,1\}^{m \times r} \text{ and } A \in \mathbb{R}^{r \times n}, \ A^\top \mathbf{1}_r = \mathbf{1}_n \text{ such that } D = TA. \tag{B.1}$$

We here provide variants of Algorithms 1 and 2 to solve the corresponding problem without the constraint $A^\top \mathbf{1}_r = \mathbf{1}_n$, that is

$$\text{find } T \in \{0,1\}^{m \times r} \text{ and } A \in \mathbb{R}^{r \times n} \text{ such that } D = TA. \tag{B.2}$$

The following Algorithm B.1 is the analog of Algorithm 1. Algorithm B.1 yields $\text{span}(D) \cap \{0,1\}^m$, which can be proved along the lines of the proof of Proposition 1 under the stronger assumption that $T$ has $r$ *linearly* independent in place of only $r$ *affinely* independent columns, which together with the assumption $\text{rank}(A) = r$ implies that also $\text{rank}(D) = r$ (cf. Section 2.1 of the paper). Algorithm B.1 results from Algorithm 1 by setting $p = 0$ and replacing $r - 1$ by $r$.

---

**Algorithm B.1** FIND VERTICES EXACT_LINEAR

1. Determine $r$ linearly independent columns $\mathcal{C}$ of $D$, obtaining $D_{:,\mathcal{C}}$ and subsequently $r$ linearly independent rows $\mathcal{R}$, obtaining $D_{\mathcal{R},\mathcal{C}} \in \mathbb{R}^{r \times r}$.
2. Form $Z = D_{:,\mathcal{C}}(D_{\mathcal{R},\mathcal{C}})^{-1} \in \mathbb{R}^{m \times r}$ and $\widehat{T} = ZB^{(r)} \in \mathbb{R}^{m \times 2^r}$, where the columns of $B^{(r)}$ correspond to the elements of $\{0,1\}^r$
3. Set $\mathcal{T} = \emptyset$. For $u = 1, \ldots, 2^r$, if $\widehat{T}_{:,u} \in \{0,1\}^m$ set $\mathcal{T} = \mathcal{T} \cup \{\widehat{T}_{:,u}\}$.
4. Return $\mathcal{T} = \{0,1\}^m \cap \text{span}(D)$.

---

The following Algorithm B.2 solves problem (B.2) given the output of Algorithm B.1.

---

**Algorithm B.2** BINARY FACTORIZATION EXACT_LINEAR

1. Obtain $\mathcal{T}$ as output from FIND VERTICES EXACT_LINEAR($D$)
2. Select $r$ linearly independent elements of $\mathcal{T}$ to be used as columns of $T$.
3. Obtain $A$ as solution of the linear system $TA = D$.
4. Return $(T, A)$ solving problem (B.2).

---

For the sake of completeness, we provide Algorithm B.3 as a counterpart to Algorithm 3 regarding the approximate case. An additional modification is necessary to eliminate the zero vector, which is always contained in $\text{span}(D)$ and hence would be returned as a column of $T$ if we used $B^{(r)}$ in place of $B_{\backslash 0}^{(r)}$ in step 2. below, whose columns correspond to the elements of $\{0,1\}^r \setminus \{0_r\}$.

---

**Algorithm B.3** FIND VERTICES APPROXIMATE_LINEAR

1. Compute $U^{(r)} \in \mathbb{R}^{m \times r}$, the left singular vectors corresponding to the $r$ largest singular values of $D$. Select $r$ linearly independent rows $\mathcal{R}$ of $U^{(r)}$, obtaining $U_{\mathcal{R},:}^{(r)} \in \mathbb{R}^{r \times r}$.
2. Form $Z = U^{(r)}(U_{\mathcal{R},:}^{(r)})^{-1}$ and $\widehat{T} = ZB_{\backslash 0}^{(r)}$.
4. Compute $\widehat{T}^{01} \in \mathbb{R}^{m \times 2^r}$: for $u = 1, \ldots, 2^r$, $i = 1, \ldots, m$, set $\widehat{T}_{i,u}^{01} = I(\widehat{T}_{i,u} > \frac{1}{2})$.
5. For $u = 1, \ldots, 2^r$, set $\delta_u = \|\widehat{T}_{:,u} - \widehat{T}_{:,u}^{01}\|_2$. Order increasingly s.t. $\delta_{u_1} \leq \ldots \leq \delta_{2^r}$.
6. Return $T = [\widehat{T}_{:,u_1}^{01} \ldots \widehat{T}_{:,u_r}^{01}]$

---

# C Matrix factorization with left and right binary factor and real-valued middle factor

We here sketch how our approach can be applied to obtain a matrix factorization considered in [1], which is of the form $TWA^\top$ with both $T$ and $A$ binary and $W$ real-valued in the exact case; the noisy case be tackled similarly with the help of Algorithm B.3 and is thus omitted.
Consider the matrix factorization problem

$$\text{find } T \in \{0,1\}^{m \times r}, \ A \in \{0,1\}^{n \times r} \text{ and } W \in \mathbb{R}^{r \times r} \text{ such that } D = TWA^\top, \tag{C.1}$$

and suppose that $\text{rank}(D) = r$. Then the following Algorithm C.1 solves problem (C.1).

---
**Algorithm C.1** THREEWAYBINARYFACTORIZATION
---
    1. Obtain $\mathcal{T}$ as output from FINDVERTICES EXACT_LINEAR($D$)
    2. Obtain $\mathcal{A}$ as output from FINDVERTICES EXACT_LINEAR($D^{\top}$)
    3. Select $r$ linearly independent elements of $\mathcal{T}$ and $\mathcal{A}$ to be used as columns of $T$ respectively $A$.
    4. Obtain $W = (T^{\top}T)^{-1}T^{\top}DA(A^{\top}A)^{-1}$.
    5. Return $(T, A, W)$ solving problem (C.1).
---

## D   Proof of Corollary 1

Corollary 1 follows directly from Proposition 1.

## E   Proof of Proposition 2

Before re-stating Proposition 2 below, let us recall problem (1) and property (4) of the paper.

$$\text{find } T \in \{0,1\}^{m \times r} \text{ and } A \in \mathbb{R}^{r \times n}, \ A^{\top}\mathbf{1}_r = \mathbf{1}_n \text{ such that } D = TA. \ (1)$$

$$\text{aff}(D) \cap \{0,1\}^m = \text{aff}(T) \cap \{0,1\}^m = \{T_{:,1}, \dots, T_{:,r}\} \ (4)$$

Let us also recall that $T$ is said to be *separable* if there exists a permutation $\Pi$ such that $\Pi T = [M; I_r]$, where $M \in \{0,1\}^{m-r \times r}$.

**Proposition 2.** *If $T$ is separable, condition (4) holds and thus problem (1) has a unique solution.*

*Proof.* We have $\text{aff}(T) \ni b \in \{0,1\}^m$ iff there exists $\lambda \in \mathbb{R}^r$, $\lambda^{\top}\mathbf{1}_r = 1$ such that

$$T\lambda = b \iff \Pi T\lambda = \Pi b \iff [M; I_r]\lambda = \Pi b.$$

Since $\Pi b \in \{0,1\}^m$, for the bottom $r$ block of the linear system to be fulfilled, it is necessary that $\lambda \in \{0,1\}^r$. The condition $\lambda^{\top}\mathbf{1}_r = 1$ then implies that $\lambda$ must be one of the $r$ canonical basis vectors of $\mathbb{R}^r$. We conclude that $\text{aff}(T) \cap \{0,1\}^m = \{T_{:,1}, \dots, T_{:,r}\}$. $\qquad\square$

## F   Proof of Theorem 1

Our proof of Theorem 1 relies on two seminal results on random $\pm 1$-matrices.

**Theorem F.1.** *[2] Let $M$ be a random $m \times r$-matrix whose entries are drawn i.i.d. from $\{-1,1\}$ each with probability $\frac{1}{2}$. There is a constant $C$ so that if $r \leq m - C$,*

$$\mathbf{P}\left(\text{span}(M) \cap \{-1,1\}^m = \{\pm M_{:,1}, \dots, \pm M_{:,r}\}\right) \geq 1 - (1+o(1))\,4\binom{r}{3}\left(\frac{3}{4}\right)^m \text{ as } m \to \infty.$$
$$\text{(F.1)}$$

**Theorem F.2.** *[3] Let $M$ be a random $m \times r$-matrix, $r \leq m$, whose entries are drawn i.i.d. from $\{-1,1\}$ each with probability $\frac{1}{2}$. Then*

$$\mathbf{P}\left(M \text{ has linearly independent columns}\right) \geq 1 - \left(\frac{3}{4} + o(1)\right)^m \text{ as } m \to \infty. \qquad \text{(F.2)}$$

We are now in position to re-state and prove Theorem 1.

**Theorem 1.** *Let $T$ be a random $m \times r$-matrix whose entries are drawn i.i.d. from $\{0,1\}$ each with probability $\frac{1}{2}$. Then, there is a constant $C$ so that if $r \leq m - C$,*

$$\mathbf{P}\left(\text{aff}(T) \cap \{0,1\}^m = \{T_{:,1}, \dots, T_{:,r}\}\right) \geq 1 - (1+o(1))\,4\binom{r}{3}\left(\frac{3}{4}\right)^m - \left(\frac{3}{4} + o(1)\right)^m \text{ as } m \to \infty.$$

*Proof.* Note that $T = \frac{1}{2}(M + \mathbf{1}_{m \times r})$, where $M$ is a random $\pm 1$-matrix as in Theorem F.1. Let $\lambda \in \mathbb{R}^r$, $\lambda^\top \mathbf{1}_r = 1$ and $b \in \{0,1\}^m$. Then

$$T\lambda = b \iff \frac{1}{2}(M\lambda + \mathbf{1}_m) = b \iff M\lambda = 2b - \mathbf{1}_m \in \{-1,1\}^m. \tag{F.3}$$

Now note that with the probability given in (F.1),

$$\text{span}(M) \cap \{-1,1\}^m = \{\pm M_{:,1}, \ldots, \pm M_{:,r}\} \implies \text{aff}(M) \cap \{-1,1\}^m \subseteq \{\pm M_{:,1}, \ldots, \pm M_{:,r}\}$$

On the other hand, with the probability given in (F.2), the columns of $M$ are linearly independent. If this is the case,

$$\text{aff}(M) \cap \{-1,1\}^m \subseteq \{\pm M_{:,1}, \ldots, \pm M_{:,r}\}$$
$$\implies \text{aff}(M) \cap \{-1,1\}^m = \{M_{:,1}, \ldots, M_{:,r}\}. \tag{F.4}$$

To verify this, first note the obvious inclusion $\text{aff}(M) \cap \{-1,1\}^m \supseteq \{M_{:,1}, \ldots, M_{:,r}\}$. Moreover, suppose by contradiction that there exists $j \in \{1, \ldots, r\}$ and $\theta \in \mathbb{R}^r$, $\theta^\top \mathbf{1}_r = 1$ such that $M\theta = -M_{:,j}$. Writing $e_j$ for the $j$-th canonical basis vector, this would imply $M(\theta + e_j) = 0$ and in turn by linear independence $\theta = -e_j$, which contradicts $\theta^\top \mathbf{1}_r = 1$.

Under the event (F.4), $M\lambda = 2b - \mathbf{1}_m$ is fulfilled iff $\lambda$ is equal to one of the canonical basis vectors and $2b - \mathbf{1}_m$ equals the corresponding column of $M$. We conclude the assertion in view of (F.3). $\square$

## G   Theorem 1: empirical evidence

It is natural to ask whether a result similar to Theorem 1 holds if the entries of $T$ are drawn from a Bernoulli distribution with parameter $p$ in $(0,1)$ sufficiently far away from the boundary points. We have conducted an experiment whose outcome suggests that the answer is positive. For this experiment, we consider the grid $\{0.01, 0.02, \ldots, 0.99\}$ for $p$ and generate random binary matrices $T \in \mathbb{R}^{m \times r}$ with $m = 500$ and $r \in \{8, 16, 24\}$ whose entries are i.i.d. Bernoulli with parameter $p$. For each value of $p$ and $r$, 100 trials are considered, and for each of these trials, we compute the number of vertices of $[0,1]^m$ contained in $\text{aff}(T)$. In Figure G.1, we report the maximum number of vertices over these trials. One observes that except for a small set of values of $p$ very close to 0 or 1, exactly $r$ vertices are returned in all trials. On the other hand, for extreme values of $p$ the number of vertices can be as large as $2^{20}$ in the worst case.

Figure G.1: Number of vertices contained in $\text{aff}(T)$ over 100 trials for $T$ drawn entry-wise from a Bernoulli distribution with parameter $p$.

## H   Entire set of experiments with synthetic data

In section 4.1 of the paper, we have presented only a subset of all synthetic data experiments that we have performed. We here present the entire set.

For the first set of experiments, we have considered three different setups concerning the generation of $T$ and $A$ and two choices of $r$ (10 and 20), out of which only the results of the first one ('T0.5') for $r = 10$ are reported in the paper.

**Setups.**

*'T0.5':* We generate $D = T^*A^* + \alpha E$, where the entries of $T^*$ are drawn i.i.d. from $\{0, 1\}$ with probability 0.5, the columns of $A$ are drawn i.i.d. uniformly from the probability simplex and the entries of $E$ are i.i.d. standard Gaussian. We let $m = 1000$, $r \in \{10, 20\}$, $n = 2r$, and let the noise level $\alpha$ vary along a grid starting from 0.

*'Tsparse+dense':* The matrix $T$ is now generated by drawing the entries of one half of the columns of $T$ i.i.d. from a Bernoulli distribution with probability 0.1 ('sparse' part), and the second half from a Bernoulli distribution with parameter 0.9 ('dense' part). The rest is as for the first setup.

*'T0.5,Adense':* As for *'T0.5'* apart from the following modification: after random generation of $A$ as above, we compute its Euclidean projection on $\{A \in \mathbb{R}_+^{r \times n} : A^\top \mathbf{1}_r = \mathbf{1}_n, \max_{k,i} A_{k,i} \leq 2/r\}$, thereby constraining the columns of $A$ to be roughly constant. With such $A$, all data points are situated near the barycentre $T\mathbf{1}_r/r$ of the simplex generated by the columns of $T$. Given that the goal is to recover vertices, this setup is hence potentially more difficult.

Figure H.1: Results of the synthetic data experiments separated according to the two setups *'T.05'* and *'Tsparse+dense'*. Bottom/top: $r = 10$, $r = 20$. Left/Middle/Right: $\|T^* - T\|_F^2/(m\,r)$, $\|T^*A^* - TA\|_F/(m\,n)^{1/2}$ and $\|TA - D\|_F/(m\,n)^{1/2}$.

Figure H.2: Results of the synthetic data experiments (continued) for the setup *'T0.5,Adense'*. Bottom/top: $r = 10$, $r = 20$. Left/Middle/Right: $\|T^* - T\|_F^2 / (m\,r)$, $\|T^* A^* - T A\|_F / (m\,n)^{1/2}$ and $\|T A - D\|_F / (m\,n)^{1/2}$.

Regarding the comparison against HOTTOPIXX, only the results for $r = 10$ are reported in the paper. We here display the results for $r = 20$ as well.

Figure H.3: Results of the experimental comparison against HOTTOPIXX.