[Reviews · NeurIPS 2013]

Submitted by Assigned_Reviewer_1

The authors present an algorithm for matrix factorization with binary components; in this particular variant the authors have a number of combinatorial (basically zero-one) constraints on the elements of one of the two factor matrices.

The paper has great theory, that was enjoyable to read. The use of the Littlewood-Offord lemma is very elegant (even though it only holds for asymptotically large values of m). The use of linear programming to further speed up the overall algorithms is also a nice approach. Overall, this is very strong paper from a theoretical perspective, certainly above the NIPS threshold.

My main concern, however, had to do with the motivation of this particular factorization. While intuitively appealing (having a zero-one matrix on the left hand size of a matrix factorization), it was not obvious to me what applications this could have. The case study described by the authors is certainly interesting, but somewhat exotic: to the best of my knowledge, associations between cancers and shifts in methylation of cell types are still vastly unexplored. I would like to see more motivation for the proposed factorization, if possible.

Overall, a solid paper that I would like to see published in NIPS.
Summary: The authors present algorithms for a matrix factorization with zero-one constraints on one of the two factor matrices. A very good paper from a theoretical perspective; however, the overall motivation for this factorization is rather weak.

Submitted by Assigned_Reviewer_3

The paper presents an algorithm for low-rank matrix factorization with constraints on one of the factors should be binary. The paper has several novel contributions for this problem. The algorithm guarantees the exact solution with the time complexity of O(mr2^r+mnr), where previous approach (E. Meeds et al., NIPS 2007) uses MCMC algorithm so that it cannot guarantee a global convergence. Under additional assumptions on the binary factor matrix T, the uniqueness of T is proved which means that each data point has a unique representation with the columns of T. Using Littlewood-Offord lemma, the paper computes a theoretical speed-up factor for their heuristic of the candidate binary vector set reduction step.
I summarize the strengths (+) and weaknesses (-) of this paper.

Quality (+) The proposed algorithm exhaustively searches all candidates of binary vectors whose time complexity is only exponential in the rank of the matrix by its algorithmic procedure, and later a heuristic of the candidate set reduction is suggested. Analysis of uniqueness and time complexity of the algorithm is appropriate to support the algorithm. (-) I’m not sure this theoretically well-supported algorithm actually works better than the previous approach (E. Meeds et al., NIPS 2007), since the paper doesn’t provide empirical comparison between them. Is this comparison is possible? As a minor comment about the algorithm, is there any reason to use the vector p in each algorithms? It seems that this vector does not contribute to the proof of theories.

Clarity (+) It is easy to follow the overall procedure of the algorithm and its theoretical justification. (-) In my opinion, the statements about a non-negative variant in the section 2.3 is obvious and unnecessary, and seems to be not connected with other contents.

Originality (+) This is a novel contribution for matrix factorization. The algorithm is completely different from the previous one, and guarantees the uniqueness of the solution. Important references about matrix factorization with binary matrices are included in the paper.

Significance: (+) Uniqueness of the solution may be useful for many problems whose representation of each data points should be unique, e. g. DNA sequence analysis, even if in most case, data has noise so that this uniqueness property for the exact case does not satisfy. The paper provides a unique theoretical behavior of matrix factorization with one binary factor matrix. (-) As I mentioned earlier, empirical comparison with the existing work is not provided, so I’m not sure the proposed algorithm is better with real-world data. But this comparison is not necessary if it is not possible, since the proposed algorithm is better enough than the existing work with only its theoretical analysis about the uniqueness of the solution.

Summary: This paper proposes a matrix factorization method with constraints on one of the factors should be binary whose time complexity is only exponential in the rank of the matrix. Theoretical analysis about the uniqueness of the solution is a significant contribution for this problem, and it would be better to add an empirical comparison with the existing work.

Submitted by Assigned_Reviewer_5

This paper discusses a new approach to binary matrix factorization
that is motivated by recent developments in non-negative matrix
factorization. The paper is well written and very clear. The goal of
the paper is to present an algorithm for finding a factorization of a
matrix in the form $D = T A$ where the entries of $T$ are in
$\{0,1\}$. Such a model has wide applicability and is of interest to
the ML community. The algorithm has provable recovery guarantees in
the case of noiseless observations. A modified algorithm is applied
to the noisy setting; however, the authors do not establish recovery
guarantees.

Can the authors provide comparisons of their method in the noisy
setting to existing methods for solving matrix
factorization problems: for example the method proposed by Bittorf
et. al. or EM (that is if we assume the latent binary factors are
independent)? Such a comparison could serve to highlight some of the
advantages of the proposed method in the noisy setting versus existing
approaches. Those methods have much better computational performance.
Can the authors also discuss under what circumstances they expect
their method to perform better than the linear programming (LP) based
method (which also has theoretical guarantees)? The method proposed in
this report exploits the information that the matrix $T$ is binary;
however, the LP based methods might also perform well in such contexts.

What is the usefullness of the discussion on uniqueness and
non-negative A? From my reading of Proposition 2, if problem (1) has a
unique solution $(T^*,A^*)$ and if those unique solutions satisfy $A^*
\geq 0$, then Algorithm 1 will return $T^*$ and $A^*$. Doesn't
Proposition (2) simply follow from Proposition (1) if both equations
(6) and (7) are satisfied? If the point is that simple, I do not see
the need to discuss it in such detail. Otherwise, could the authors
please clarify that point.
Summary: Overall the paper is interesting and well written. However, adding
deeper comparisons to the existing work in matrix factorizations and
NMP will add better insight into the behavior of the algorithm and the
main contributions of this article.
Author Feedback

Author rebuttal: We thank all reviewers for their constructive and helpful comments.

R1: 'There is little motivation for the particular factorization discussed in the paper. The case study, an application to DNA methylation array data, is somewhat exotic.'

Analysis of methylation array data has in fact received little attention outside the bio-medical literature, where it is an important, emerging topic (pubmed yields more than 15000 hits for “DNA methylation AND cancer”). Advanced analysis of this kind of data is still in its infancy, and we feel that the matrix factorization of the paper offers a valuable contribution in this regard.
Apart from that, the factorization can potentially be applied to other problems in biology, where “on/off phenomena” play a role (e.g. transcription factor binding, presence/absence of SNPs), as well as to blind source separation as mentioned in our introduction along with supporting references.


R2/R3: 'To better judge the performance of the proposed method, it would be useful to include comparisons against existing approaches to the problem such as [Meeds et al., NIPS 2007] and [Bittdorf et al., NIPS 2012]'

We have meanwhile conducted an experimental comparison to the LP approach of Bittdorf et al. in the separable setting with binary T. We have found that our approach is more robust to noise, and we plan to add these results in the final version.
The model of Meeds et al. involves two binary factors in a three-factor factorization and is hence different from our factorization model. A comparison can still be performed when running our approach in a two-step manner. Provided that code can be obtained from Meeds et al, such a comparison would be included in the final version.
Please note that the paper already contains a comparison to two methods based on alternating optimization (a standard approach to NMF), adapted to our specific factorization problem.


R2: 'It is worth discussing in which settings the authors' method performs better than the LP method of Bittdorf et al.'

The LP method does not require the left factor T be binary, and it has no more than polynomial complexity in the rank r of the factorization. On the other hand, (approximate) separability is crucial to the LP approach – all theoretical results are limited to this specific setting (even though one may argue that separability is satisfied in major applications of NMF). Moreover, in the noisy case, the LP approach needs suitable specification of a second tuning parameter (depending on the noise level) besides the rank r.
We plan to include these points along with the results of the empirical comparison in the final version.


R2/R3: 'The statement (Proposition 2) about a variant with non-negative A appears to be unnecessary and obvious in light of Proposition 1 on the general case'

We agree that the statement is obvious. Nevertheless, we believe that it deserves to be stated (possibly more adequately as corollary to Proposition 1 in a revised version), because 1) a modification to our algorithmic approach introduced earlier for general A would be necessary without uniqueness 2) the application presented in the last section requires non-negative A.


R2: 'Is there any reason to use the vector p in the algorithms/theory ? It seems that this vector is not essential.'

The vector p in aff(D) is necessary as we work in the affine hull of D instead of its linear hull.
However, similar results would be possible for the linear hull (dropping the sum-to-one constraints on A), in which case one would omit p.